# Exploration of Contributory Factors to an Unpleasant Bracing Experience of Adolescent Idiopathic Scoliosis Patients a Quantitative and Qualitative Research

**DOI:** 10.3390/children9050635

**Published:** 2022-04-28

**Authors:** Huan Wang, Xianglong Meng, Daniel Tetteroo, Frank Delbressine, Yaozhong Xing, Keita Ito, Yong Hai, Panos Markopoulos

**Affiliations:** 1Department of Industrial Design, Eindhoven University of Technology, 5612 AP Eindhoven, The Netherlands; d.tetteroo@tue.nl (D.T.); fdelbres@tue.nl (F.D.); p.markopoulos@tue.nl (P.M.); 2Department of Orthopaedic Surgery, Beijing Chaoyang Hospital, Beijing 100020, China; mengxianglong99@126.com (X.M.); xingyaozhong@ccmu.edu.cn (Y.X.); 3Academic Department of Orthopaedic Surgery, Capital Medical University, Beijing 100069, China; 4Department of Biomedical Engineering, Eindhoven University of Technology, 5612 AP Eindhoven, The Netherlands; k.ito@tue.nl

**Keywords:** adolescent idiopathic scoliosis, brace treatment, correction force, discomfort level

## Abstract

Background: To obtain a better understanding of the wearing habits and preferences of Adolescent Idiopathic Scoliosis (AIS) patients undergoing rigid brace treatment, we examine what factors contribute to patients’ perceived discomfort during the treatment. Methods: Seventeen AIS patients treated with a rigid brace were recruited. We asked them to complete a questionnaire and participate in an interview study. Finally, we measure the interface corrective force and perceived discomfort with the participants for different positions and assess the correlation. Results: Our survey reveals that participants scored the lowest in the domains of environmental factors, psycho-spiritual factors, satisfaction, and self-image. Appearance anxiety, physical and psychological discomfort and inconvenience were the three most frequently mentioned problems in the interviews on participants’ daily bracing experiences. A significant, moderately positive relationship between corrective force and discomfort level was found only when participants were lying on their left side, but not in any of the other positions. No significant correlation between treatment length and perceived discomfort was found. Conclusions: Future work should focus on reducing the psychological burden and the inconvenience of wearing a brace, rather than on reducing physical discomfort resulting from the corrective force.

## 1. Introduction

Adolescent Idiopathic Scoliosis (AIS) is defined as a three-dimensional spinal condition with unusual twisting curvature, which normally happens amongst children. It can be divided into early onset scoliosis (below the age of 10 years) or adolescent scoliosis (above the age of 10 years) [1]. The International Scientific Society on Scoliosis Orthopaedic and Rehabilitation (SOSORT) guidelines recommend observation and exercise for curvatures less than, or equal to 20°. For moderate scoliosis with curvatures between 20–35°, brace treatment is suggested. Bracing is the most effective option to prevent curvature progression of pre-adolescents with moderate scoliosis (curvature exceeding 30°) while for patients with severe scoliosis (curvature more than 45–50°), surgical treatment is deemed necessary [1,2].

Rigid bracing has been the most common non-operative treatment for the prevention of curvature progression and avoidance of surgery [1,2,3,4], though bracing may impair respiratory mechanics, resulting in a reduction of pulmonary function [5,6,7,8,9], or cause pressure scars, pain, emotional and social problems, a disturbed self-image and body image, thereby affecting the quality of life [10]. These unfavorable impacts on patients’ physical and psychological discomfort, in some instances even pain are reported to be associated with low compliance [11,12,13]. According to the Comfort Theory, patients will engage with more attention in seeking healthy behaviors, and treatments tend to have more positive outcomes if patients are more comfortable [14,15]. In our study, we follow the rationale behind this theory in measuring the perceived comfort levels for scoliotic adolescents under brace treatment, to pursue a better understanding of the factors contributing to perceived discomfort, and to obtain insights into how treatment outcomes could be improved by improving the wearing comfort. We hypothesize that perceived discomfort might be related to physical aspects, such as interface corrective force and treatment length, or to psychological aspects, such as mental health, self-image and social anxiety. This study aims to identify the main causes of an unpleasant bracing experience. We explore the relationship between discomfort level and corrective force in different positions, such as standing, sitting, and lying down; the relationship between discomfort level and treatment duration to understand if participants would become accustomed to bracing as treatment length increases and whether perceived discomfort is related to psychological factors, such as mental health, self-image, and social anxiety.

## 2. Materials and Methods

### 2.1. Participants

Seventeen adolescent patients were recruited following the SOSORT guidelines [2] and criteria proposed by Richards et al. [16]. All participants were fully capable of reading and writing, aged between 10 and 17 years, and were undergoing rigid brace treatment, attending the scoliosis clinic of Beijing Chaoyang Hospital for regular visits between October 2019 to February 2020. All patients and their parents gave written informed consent to the measurement of the exerted corrective force and the comfort level during bracing and they were also informed that they could withdraw at any time. Data were collected at the clinic. 

The study was approved by the Ethical Committee of Beijing Chaoyang Hospital and the Ethical Review Board at the Eindhoven University of Technology (No. ERB2019ID8).

All participants were diagnosed with AIS and under treatment with an adjusted Cheneau brace or a Boston brace, as shown in Figure 1. This adjusted Cheneau brace is designed by Rodin4D [17] based on a scan of the patient’s body and manufactured using 3D printing technology. The Boston brace is a prefabricated polypropylene pelvic module with a soft foam polyethylene lining, adjusted to the patient’s needs with lumbar and thoracic pressure pads. Table 1 lists the characteristics of all 17 participants and the treatment length of each participant. Treatment length refers to the elapsed time between the time participants had their initial brace assessment, which was designated as month 0.

### 2.2. Measures

Three questionnaires and one interview were used to examine patients’ psychological status during brace treatment. Perceived discomfort and interface corrective force under eight positions (standing, sitting, supine, prone, standing with a single leg, lying on one side) were recorded to explore the relationship between perceived discomfort and interface corrective force.

#### 2.2.1. Questionnaires

The Scoliosis Research Society-22 questionnaire (SRS-22), General Comfort Questionnaire (GCQ), and Oswestry Disability Index (ODI) were administered to the participants at the start of the study. The SRS-22 and GCQ were used to evaluate patients’ quality of life and comfort levels. The ODI was initiated by John O’Brien in 1976 and has been used to measure the lower back pain of patients with spine disorders in activities of daily living, as our study also explores the comfort level during eight different positions to simulate daily life, the ODI test was used to check if the discomfort is from lower back pain caused by spine conditions.

The Scoliosis Research Society HRQL (health-related quality-of-life) questionnaire was developed by Hasher [18] to provide an assessment questionnaire for patients with idiopathic scoliosis. The SRS-22 questionnaire measures five domains of Quality of Life: mental health, self-image, function/activity, pain and satisfaction. Each domain is graded from 1 (worst) to 5 (best). In this study, the Chinese version [19] of SRS-22 was administered to the participants.

Lorente et al. [20] conducted a review of the instruments to assess patient comfort and found that General Comfort Questionnaire (GCQ) is an adequate instrument for assessing the comfort level of patients. GCQ [21] is a generic, self-report instrument developed to assess a patient’s comfort. It consists of 48 4-point Likert scales (from strongly agree to strongly disagree) items covering physical, spiritual, environmental, and social dimensions. Considering the age of our participants ranged from 10 to 17 years old, and to avoid participants scoring on medium quarters, we converted the 4-point Likert scale into a 10-point Likert scale to assess patients’ comfort. Two independent translators (one with a medical background) converted the original English texts into Chinese. A pilot study involving six participants was conducted to analyze the internal consistency of the Chinese-GCQ, which was found to have a Cronbach’s alpha coefficient of 0.713, indicating that the internal consistency of Chinese-GCQ is acceptable. 

The scores of the SRS-22 and the GCQ were mapped to a 0 or 1 scale for comparability.

The Oswestry Disability Index (ODI) [22] measures disability levels through 10 items with six statements scored from 0 to 5. The Chinese version of ODI [23] was administered to the participants. The disability percentage is calculated as a percentage of the total achievable score. The interpretation of ODI scores is as follows:
0–20%: minimal disability: The patient can cope with most living activities. Usually, no treatment is indicated apart from advice on lifting, sitting and exercises.21–40%: moderate disability: The patient experiences more pain and difficulty with sitting, lifting, and standing. Travel and social life are more difficult, and they may be disabled from work. Personal care, sexual activity and sleeping are not grossly affected, and the patient can usually be managed by conservative means.41–60%: severe disability: Pain remains the main problem in this group, but activities of daily living are affected. These patients require a detailed investigation.61–80%: crippled: Back pain impinges on all aspects of the patient’s life. Positive intervention is required.81–100%: These patients are either bed-bound or exaggerating their symptoms.

#### 2.2.2. Interface Corrective Force Measurement

To perform body/brace interface corrective force measurements, we used the Tekscan FlexiForce Electronic (OEM Development Kit), manufactured by TekScan (Boston, MA, USA). The kit gathers analog data, produces immediate force measurements and records at a frequency of 10 Hz. A FlexiForce sensor 502 was implemented with a double-sided tape at the inner side of the brace to measure the interface corrective force exerting on the body, as shown in Figure 2. The sensor was placed on the primary curve of the brace. The location of the sensor was determined by a physician based on the X-ray pictures of the participants.

#### 2.2.3. Discomfort Level

A feeling-of-discomfort slider, as shown in Figure 3, was used to enable participants to indicate their level of discomfort throughout the test. The slider is based on research by Walker et al. [24], who designed a motorized slide-potentiometer connected to a tablet computer using an Arduino Uno, to investigate participants’ continuous behavior [25]. The Arduino was programmed to take a reading of the current value of the potentiometer at 10 Hz. The extremities of the potentiometer were mapped to 0 and 100, the endpoints corresponding to “Comfortable” and “Extremely Uncomfortable”, respectively.

Discomfort level and interface corrective force were measured for 30 s, resulting in 300 data points per participant and position. We then averaged these 300 data points per participant and position, resulting in a total of 122 data points (we obtained fewer than 136 data points (17 × 8), since seven participants refused measurements in laid down positions because of expected discomfort). Pearson’s correlation coefficient was calculated using SPSS (SPSS Inc., Chicago, IL, USA) to evaluate the strength and direction of the linear relationship between discomfort level and interface corrective force for all participants.

#### 2.2.4. Interview Survey

To survey patients’ wearing habits and expectations of their brace, a series of open questions were asked, as shown in Table 2. Three out of 17 participants refused participation in the interview because of personal reasons. Interviews were conducted by Huan wang. The interview lasted between 15–30 min. All interviews were audio-recorded and transcribed. The NVivo software platform [26] was used to perform a content analysis of the interview results to explore the most frequently mentioned concerns. Word clouds were created to show patients’ primary concerns on bracing experience.

### 2.3. Study Procedure

Participants first were requested to complete the questionnaires. Then, the researchers attached the sensor for measuring interface corrective force to the participant’s brace.

After positioning the sensor between the brace and the body, we measured the corrective forces in eight different positions: standing, sitting, supine, prone, standing on one leg (left and right), lying on one side (left and right), aiming to cover daily wearing positions [27]. Simultaneously, discomfort levels were indicated by participants. Each position lasted for 30 s.

Finally, the participants were invited to participate in the closing interview.

## 3. Results

Sixteen out of 17 participants gave the complete results of the questionnaires survey; seven out of 17 participants refused measurements in laid down positions because of expected discomfort all participants, so we missed corrective force and discomfort level in two laying down positions from these seven participants; three out of 17 participants refused participation in the interview because of personal reasons.

### 3.1. Questionnaires

The results of the questionnaires are shown in Table 3. Figure 4 visualizes the scores of the SRS-22 and the GCQ using Box-whisker plots. In the GCQ questionnaire, participants scored lowest on the environmental (mean = 2.7, SD = 1.0) and the psycho-spiritual (mean = 2.9, SD = 0.94) domains, with the psycho-spiritual domain showing a slightly smaller variation. Through the SRS-22 questionnaire, participants reported lowest scores for satisfaction (mean = 3.0, SD = 0) and self-image (mean = 3.0, SD = 0).

These results indicate that participants undergoing brace treatment have concerns about external factors and circumstances. ƒ These concerns exacerbate participants’ feelings about their living environment, including light, noise, color, temperature, the safety of the environment and the landscape visible through the window, as revealed from the environmental dimension within the GCQ survey. Moreover, participants indicated difficulties in appearance identification and dissatisfaction with treatment management through the SRS-22 survey.

According to outcomes of the ODI, 12 out of 17 (71%) participants have minimal disability and are capable of most living activities. Five out of 17 (29%) participants have moderate disability and are experiencing more pain and difficulty with sitting, lifting, and standing, while personal care and sleeping are not gravely affected.

### 3.2. Discomfort Level and Interface Corrective Force

Pearson’s correlation coefficient for the discomfort level and the interface corrective force are shown in Table 4. No significant relation between interface corrective force and perceived discomfort was found for any of the conditions, except a significant moderate positive relationship (*r* = 0.673, *p* = 0.033) for the position of lying on the left side. 

### 3.3. Treatment Length and Perceived Discomfort

We also examined the correlation between treatment length and perceived discomfort for the different positions. Table 5 shows the Pearson correlation coefficients between the treatment length and the discomfort level for the eight different positions. No significant correlation was found.

### 3.4. Effect of Position on Perceived Discomfort

To observe if there is a most uncomfortable position in participants’ daily bracing experience, we analyzed the distribution of reported discomfort levels for each position. According to the recommended cut points on the pain Visual Analogue Scale [28,29] which is based on a similar measuring principle as the feeling-of-discomfort slider, we described the discomfort intensity of the participants as follows: no discomfort (0–4 mm), mild discomfort (5–44 mm), moderate discomfort (45–74 mm), and severe discomfort (75–100 mm). Table 6 and Figure 5 show the distribution of the discomfort intensity of 17 participants in all positions. On average, participants experienced mild discomfort in all positions. The most uncomfortable position was the prone position (mean = 24.97, SD = 36.67), however, based on the large standard deviation, we are cautious to conclude that, in fact, this is an experience shared across the study population.

### 3.5. Interview Survey

The results of the interview analysis are shown in Figure 6. Appearance and physical and psychological discomfort were the two most frequently mentioned problems in daily brace treatment. Nine participants (64%) preferred wearing the brace during bedtime, and seven participants (50%) deemed their body shape as “too bad, ugly, abnormal”. When asked for their primary concern regarding wearing a brace, five participants (42%) mentioned mal-appearance, while four participants (33%) named inconvenience and three participants (25%) named discomfort.

Two out of 14 participants deemed both inconvenience and mal-appearance as their top concerns and were the primary reasons for their nonadherence. Participant #15 told us: *“I’ve left school since the second month of my treatment, because it’s not convenient to put on and take off the brace at school without help, and it’s ugly. I feel bad about my body shape wearing the brace. I care about appearance. Now I wear the brace for about 20 h/day at home, but it won’t be so long after I go back to school next semester.”* She refused to wear the brace in the initial treatment because of pain and a bulge on the backside of the brace, with the hope of having surgery, which would be short-term and without impact on body appearance from her point of view. After a one-week-long persuasion by her parents, she agreed to wear the brace, but surgery would still be her first choice if no positive results would be obtained from wearing the brace within six months. Participant #4 was experiencing similar concerns, he said: *“I only wear the brace at home. It’s extremely inconvenient to put on or take off the brace without help at school and it’s visible and cannot be hidden under my clothes, especially in summer. My appearance with the brace kept bothering me a lot since the beginning of my brace treatment. I felt pain in the beginning of my treatment, and I still feel pain, but I can stand it now, while the concerns on the inconvenience and mal-appearance do not change.”* Based on the interviews, these two participants have been experiencing much more psychological disturbances than any other participants, which directly resulted in low adherence. 

## 4. Discussion

To our knowledge, this study presents the first evaluation of the continuous discomfort level in scoliosis bracing, while exploring the correlation between discomfort level and interface corrective force under daily routines of bracing patients. We hypothesized that the discomfort level in different positions, such as standing, sitting, and lying down is related to the interface corrective force. More precisely, we hypothesized there might be a positive relationship between discomfort level and corrective force. However, our results provide no evidence to confirm this hypothesis. The interface corrective force varied between different participants and different positions, similarly, mixed findings are reported in the literature, where Van den Hout et al. [27] measured the direct compressive forces exerted by the pads in Boston brace from 16 patients and found the highest pressure from the lumbar pad in the supine position. However, Ahmad et al. [30] analyzed the interface pressure exerted by the Cheneau brace in 72 patients and found the highest pressure in the position of lying right side. 

Regarding the correlations between interface corrective force and perceived discomfort under eight different positions, only a significant moderate positive relationship was found in the position of lying on the left side, though neither interface corrective force nor discomfort level in this position was the highest of all eight positions. We were unable to explain the difference with the insignificant relationship in the position of lying on the right side. We hypothesize that the correlation found might be accidental rather than systemic and that in general there is no strong correlation between interface corrective force and discomfort. There is no agreement on whether brace treatment could cause discomfort or what are these discomfort related to. Some studies [31,32] proved that patients undergoing brace treatment are experiencing feelings of shame and physical discomfort, while some researchers believed that uncomfortable braces generally are a result of poor workmanship [33].

If we turn toward the results from the questionnaires and the interview, we find an interesting interpretation of the force and discomfort measurement results. Participants indicated difficulties in appearance identification through the SRS-22 questionnaire, and the interview data show that bedtime was the preferred wearing time for 64% of the participants, because they are disturbed by negative peer attitudes, the difficulty of wearing the brace at school without help, or because of critical comments about their physical shape during daytime wearing. Apparently, these participants prefer withstanding the mild discomfort of wearing a brace during nighttime over an unpleasant bracing experience during daytime. Auerbach et al. [34] used the SRS-22 questionnaire to access scoliotic patients’ body image disturbance and patients scored the lowest in the self-image domain, which is in concordance with our results, and other studies [35,36]. Other studies using the ODI questionnaire also showed that participants have a minimal disability and are capable of most living activities, for instance, Bayrak et al. [37] reported 13/100, and Lange et al. [38] reported 8/100, compared to our results of 8.8/100. Leszczewska et al. [39] found that patients treated with brace suffer from stress more than patients with physiotherapy. Similar results were reported by a review paper [7] which showed that patients undergoing brace treatment are experiencing medium stress caused by limited physical movement and unpleasant body configuration. Psychological burden other than corrective force seems to be the main contributory factor to an unpleasant bracing experience in these cases.

Another correlation that we investigated is between treatment length and perceived discomfort. Van den Hout et al. [27] reported no statistically relevant difference between the corrective forces in a new (<6 months) and old (>6 months) brace. Therefore, we assumed that the corrective forces generated by new and old braces are the same. However, we also hypothesized that there might be a negative correlation between perceived discomfort and treatment length. More precisely, as treatment length increases, discomfort levels might decrease because participants become accustomed to bracing. However, in our study, no significant correlation was found. The discomfort level showed no significant decrease in participants with a treatment length of more than 6 months, compared to the participants with a treatment length of fewer than 6 months. 

The survey and interview results clearly revealed participants’ concerns about their physical shape. Lower scores for self-image and satisfaction with treatment management were observed from the results of the SRS-22 questionnaire, and mal-appearance was the most frequently mentioned concern during the interviews, regardless of the treatment length of a participant. These findings confirm previous studies that have shown poorer self-image caused by a rigid brace, negative peer attitudes, and critical comments about a patient’s physical shape may contribute to a negative body configuration, especially among female patients [40,41,42]. Law et al. [43] suggested that an aesthetically pleasing brace and the involvement of patients in the design process of the brace are important for increasing user compliance. 

In our study, inconvenience was also reported as the primary concern by four participants (33%) using the Boston and the Cheneau braces, and in our interviews, eight participants (six in the Cheneau brace and two in the Boston brace) mentioned that, at school, it is not easy to put the brace on and take off without help. The Boston brace is made of high-density polypropylene lined with polyethylene foam and with a posterior opening using Velcro straps, which are not reachable and adjustable for patients themselves. In contrast, the Cheneau brace is designed as an anterior closing with Velcro straps which allow for independence. However, both participants wearing the Boston brace as well as the Cheneau brace reported inconvenience. It is important to note this concern, although we do not have solid evidence to show that this inconvenience itself causes low adherence. 

Concluding, to improve bracing outcomes of AIS patients, we need to understand patients’ bracing experiences and primary concerns. Based on the outcomes of our study, an unpleasant bracing experience seems to be caused mainly by psychological burden rather than interface corrective force. We believe more future work on patients’ body configuration, mental health, and the identification of themselves as brace wearers may help researchers understand patients’ bracing experience, the data from this study could be a reference for further studies with a large sample of participants. Meanwhile, we think that in future work the relationships between patients’ curve type, brace type, cobb angle and comfort level are also worth to be explored. A more accurate daily track of interface force or pressure, for instance, obtained with monitors attached to the brace, would provide an overview of biomechanical parameters, interface forces we measured from different positions could be a reference.

## 5. Limitations

Our study has some limitations. Our sample is relatively small, so our data should be interpreted cautiously. Further studies with larger sample sizes are needed to verify our conclusions. Such studies would also be useful to verify whether results vary based on the brace type. Finally, the discomfort level experienced during a short interval (30 s for each position) may not be representative of the discomfort level experienced in the daily bracing experience and could not provide sufficient information to identify the most uncomfortable position. Nonetheless, taken together with the results from the questionnaires and interviews, we argue that there is convincing evidence that discomfort in bracing is at least to a large extent dependent on factors beyond the brace’s corrective force.

## 6. Conclusions

Our results suggest that interface corrective force is not the major contributory factor to an uncomfortable bracing experience. The bracing discomfort is more likely caused by participants’ serious concerns regarding their self-image and psycho-spiritual difficulties caused by negative peer attitudes and critical comments about their physical shape.

Future work should focus on reducing the psychological burden and the inconvenience of wearing a brace, rather than on reducing physical discomfort resulting from the corrective force.

## Figures and Tables

**Figure 1 children-09-00635-f001:**
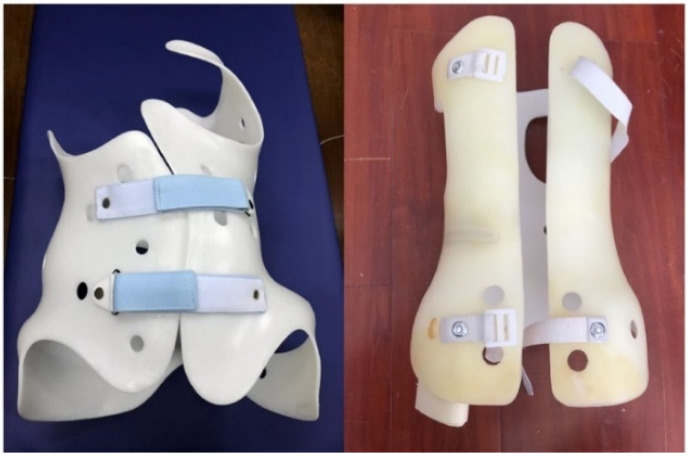
Adjusted Cheneau brace (**left**) and Boston brace (**right**) used by participants.

**Figure 2 children-09-00635-f002:**
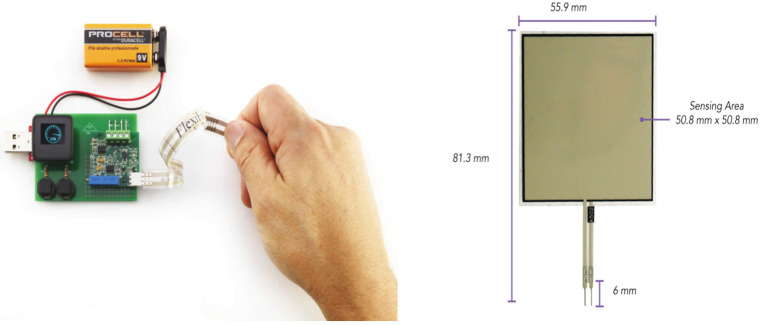
OEM Development Kit and FlexiForce sensor A502.

**Figure 3 children-09-00635-f003:**
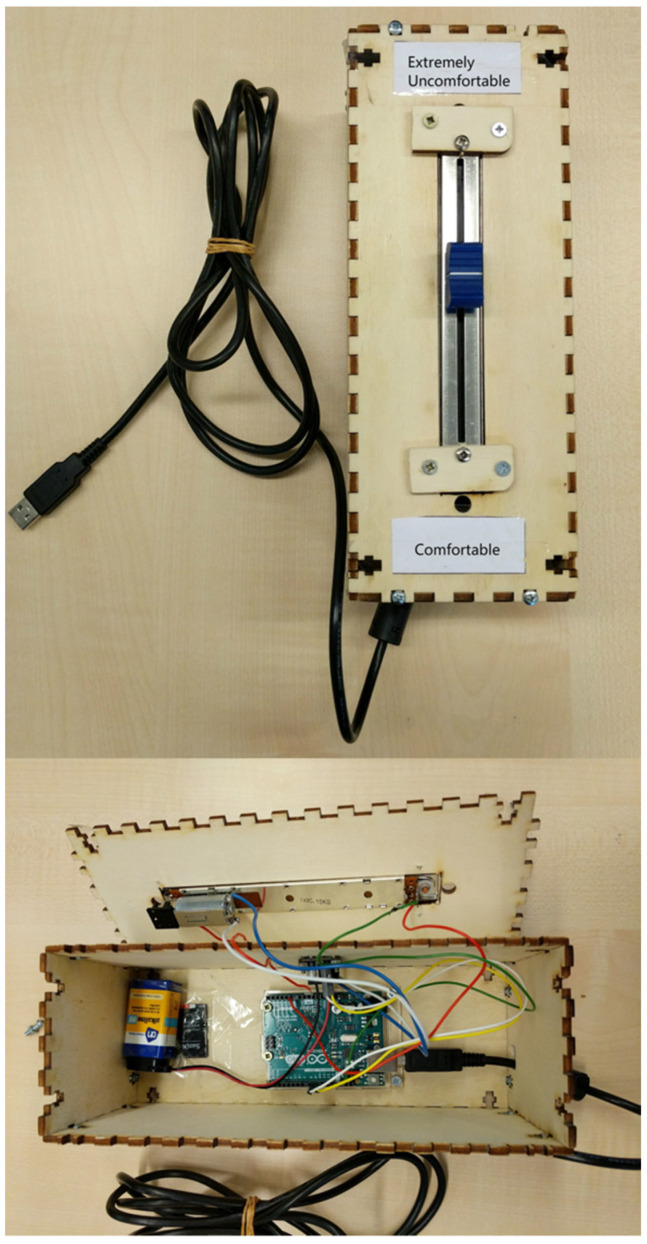
Feeling-of-uncomfortable slider.

**Figure 4 children-09-00635-f004:**
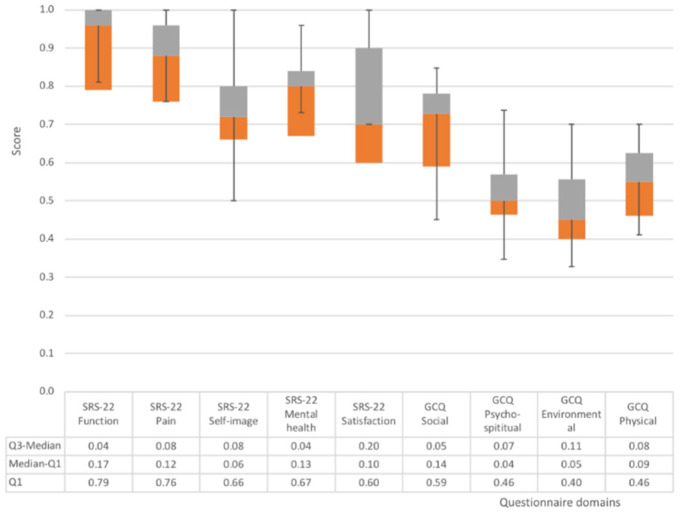
Box-whisker plot for the SRS-22 and GCQ domain scores.

**Figure 5 children-09-00635-f005:**
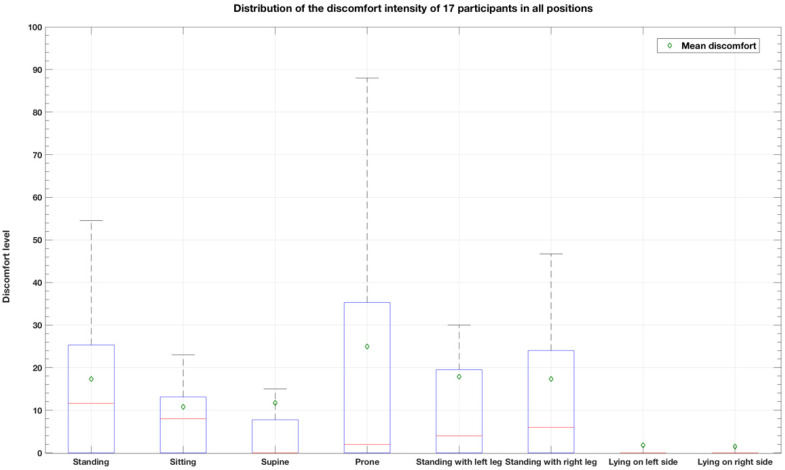
Distribution of the discomfort intensity of 17 participants in all positions.

**Figure 6 children-09-00635-f006:**
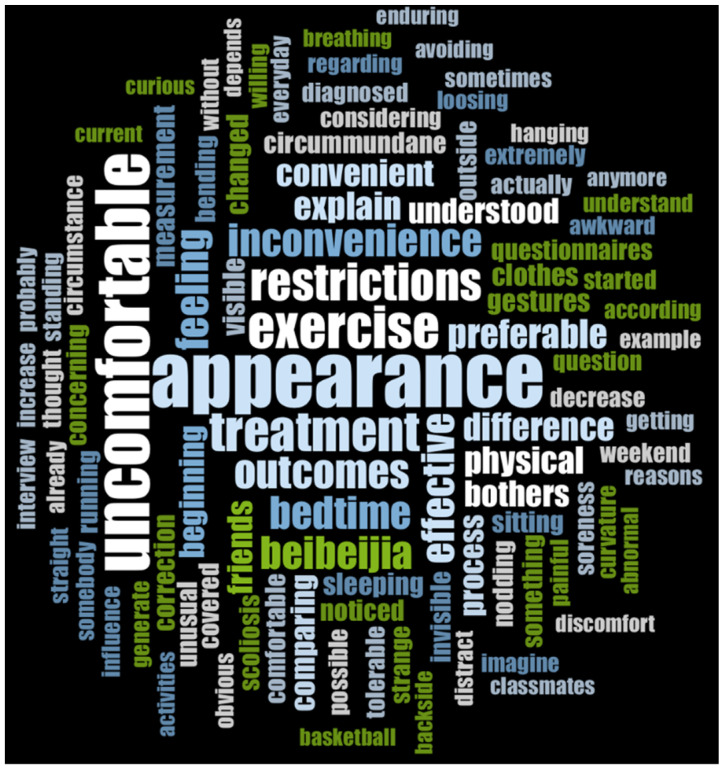
Content analysis on Interview survey.

**Table 1 children-09-00635-t001:** Characteristics of the 17 participants.

No.	Sex	Age [Years]	Treatment Length [Months]	AIS Curve Type	Cobb Angle [°]	Apex
1	Female	11	0	Right thoracic/left lumbar	35.4/21.1	T10/L4
2	Female	15	0	Right thoracic/left lumbar	34.2/37.1	T9/L3
3	Female	14	2	Right thoracic/left thoracic lumbar	30.4/25.5	T8/L2
4	Male	15	5	Right thoracic/left thoracic lumbar	23.8/24.8	T8/L1
5	Female	12	3	Right thoracic/left thoracic lumbar	40.8/30.1	T7/T12
6	Male	16	0	Right thoracic/left lumbar	28.8/31.1	T8/L2
7	Female	12	4	Right thoracic/left thoracic lumbar	12.8/26.5	T5/L3
8	Male	17	11	Right thoracic lumbar/left lumbar	23.1/25.4	T12/T5
9	Female	11	2	Right thoracic/left lumbar	36.5/23.8	T10/L4
10	Male	10	72	Left thoracic lumbar/right lumbar	24.4/27.7	TL11/L4
11	Female	13	0	Left thoracic/right lumbar	18.6/25.1	T9/L2
12	Female	16	0	Right thoracic/left lumbar	39.6/21.4	T9/L3
13	Female	13	0	Left thoracic/right lumbar	22.3/43.9	T9/L2
14	Female	11	0	Right thoracic/left lumbar	40.6/27.6	T9/L3
15	Female	15	3	Right thoracic/left lumbar	40/35	T9/L3
16	Female	11	25	Right thoracic/left lumbar	47.8/43.3	T8/T11
17	Female	14	5	Right thoracic/left lumbar	19.7/14.9	T8/T2

**Table 2 children-09-00635-t002:** Open questions list in the interview phase.

No.	Questions
1	Preferable wearing time? (day time, school time, bedtime, etc.)
2	Feelings about body appearance?
3	Solutions when you were feeling extremely uncomfortable? (Loosening strap, readjusting body position, etc.)
4	Make an order of these three concerns: Uncomfortable, Malappearance, Inconvenience?
5	Any expectations on the brace design?
6	Do exercises or not?

**Table 3 children-09-00635-t003:** Participants’ domain scores at the SRS-22, GCQ and ODI.

Participant No.	SRS-22					GCQ				ODI
	Function/Activity	Pain	Self-Image	Mental Health	Satisfaction	Social	Psycho-Spiritual	Environmental	Physical	
1	3.2	4.4	3.0	4.2	3.5	1.9	1.9	1.9	1.9	33%
2	3.4	3.2	3.0	3.2	3.0	3.5	3.5	3.9	4.1	22%
3	5.0	4.4	3.8	3.2	3.5	4.8	3.0	2.8	4.2	2%
4	4.0	4.0	3.0	3.8	3.0	3.6	3.4	2.8	2.8	22%
5	4.8	4.8	3.6	3.4	4.5	4.7	2.8	2.6	3.5	0%
6	5.0	5.0	4.0	3.4	4.5	4.5	4.4	3.9	4.1	0%
7	4.2	3.6	3.6	4.2	3.0	3.6	2.9	2.4	2.8	24%
8	5.0	5.0	3.4	4.2	3.0	4.5	3.2	2.4	3.7	0%
9	3.8	3.8	4.0	4.2	3.5	4.7	2.8	2.6	3.5	0%
10										0%
11	5.0	4.4	3.6	3.0	3.0	2.7	2.8	2.9	3.2	2%
12	4.8	3.8	3.6	4.0	4.0	4.6	3.7	3.3	2.6	2%
13	5.0	4.8	3.4	4.4	5.0	4.1	2.7	2.5	2.9	9%
14	5.0	5.0	4.2	4.8	5.0	4.2	3.3	3.5	3.8	0%
15	3.4	3.6	2.2	3.0	3.0	3.4	3.7	4.2	3.4	22%
16	4.6	4.8	5.0	4.6	5.0	5.1	3.0	1.8	2.8	9%
17	5.0	4.8	4.4	4.0	4.0	4.9	2.5	1.7	3.2	0%
Mean	3.7	3.6	3.0	3.5	3.0	3.8	2.9	2.7	3.1	
SD	0.42	0.57	0	0.42	0	1.29	0.94	1.0	1.0	

SD: Standard Deviation. SRS-22 scale: 5 = best; 1 = worst; GCQ scale: 4 = best; 1 = worst; SRS-22 and GCQ questionnaire survey from Participant #10 missed.

**Table 4 children-09-00635-t004:** Correlation between interface corrective force and discomfort level for all positions.

Position	Pearson Correlation	Sig0. (2-Tailed)	No.
Standing	−0.092	0.725	17
Sitting	−0.037	0.887	17
Supine	−0.045	0.864	17
Prone	0.001	0.996	17
Standing with left leg	−0.167	0.521	17
Standing with right leg	−0.098	0.707	17
Lying on right side	0.508	0.134	10
Lying on left side	0.673 *	0.033	10

* Correlations is significant at the 0.05 level (2-tailed).

**Table 5 children-09-00635-t005:** Correlation between treatment length and discomfort level for 8 different positions.

Positions	Treatment Length		N
	Pearson *r*	Sig0. (2-Tailed)	
Standing DL	−0.350	0.169	17
Sitting DL	−0.319	0.213	17
Supine DL	−0.188	0.470	17
Prone DL	−0.286	0.266	17
Standing with left leg DL	−0.274	0.288	17
Standing with right leg DL	−0.228	0.379	17
Lying on left side DL	−0.063	0.863	10
Lying on right side DL	−0.121	0.739	10

**Table 6 children-09-00635-t006:** The discomfort intensity of 17 participants in all positions.

Positions	Participants N.	Discomfort Level			
		Minimum	Maximum	Mean	Std. Deviation
Standing	17	0.00	69.00	17.30	20.30
Sitting	17	0.00	68.50	10.81	16.65
Supine	17	0.00	100.00	11.69	26.99
Prone	17	0.00	100.00	24.97	36.67
Standing with left leg	17	0.00	82.00	17.87	28.62
Standing with right leg	17	0.00	75.00	17.33	25.26
Lying on left side	10	0.00	18.00	3.03	5.86
Lying on right side	10	0	19	2.50	6.01

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
