# Peer review of "Exploration of Contributory Factors to an Unpleasant Bracing Experience of Adolescent Idiopathic Scoliosis Patients a Quantitative and Qualitative Research"

_children, 2022, doi:10.3390/children9050635_

Round 1

Reviewer 1 Report

This manuscript aimed to identify the main causes of an unpleasant bracing experience. Seventeen AIS patients treated with rigid brace were recruited. The authors asked them to complete a questionnaire and participate in an interview study. Finally, the authors measure the interface corrective force and perceived discomfort with the participants for different positions and assess the correlation. In this survey reveals that participants scored the lowest on the domains of environmental factors, psycho-spiritual factors, satisfaction, and self-image.

I read the article with interest, the title is well thought out and faithfully reflects the content of the study, although it would be appropriate to specify the characteristics of the study to be clearer to the reader

 The abstract is adequately developed.                                                                                                                                                                                       

In the introduction, the characteristics of Idiopathic scoliosis have been shortly described. The discussion is sufficiently developed, even if a little too synthetic.

Comment 1: In the abstract: It would be appropriate to apply it, adding more information and dividing it into paragraphs (background, material and methods, results and conclusion) to make it clearer to the reader the characteristics, aim and the design of the study.

Comment 2: In the introduction: “Adolescent Idiopathic Scoliosis (AIS) is defined as a three-dimensional spinal condition with unusual twisting curvature, which normally happens amongst children. It can be divided into early onset scoliosis (below age of 10 years) or adolescent scoliosis (above the age of 10 years). Currently, brace treatment (for curvatures below 20-45°) and surgical treatment (for curvatures above 45°) are the two main methods for halting the progression of the curvature.” Please add suitable bibliographic references.

Comment 3: In the introduction: further features on diagnosis and treatment should be briefly described inserting some references, for example (Di Maria F, et al. (2021) "Immediate Effects of Sforzesco® Bracing on Respiratory Function in Adolescents with Idiopathic Scoliosis. Healthcare (Basel))".

Comment 4: In the discussion: It would be appropriate to describe what could be done in future studies to improve these preliminary data.

Comment 5: Finally, additional English editing is needed. The Non-Native Speakers of English Editing Certificate was not signed.

Reviewer 2 Report

Dear Authors, 

a well written and interesting manuscript. Only a few details to mention/add:

Introduction:

1. As you talk about moderate AIS it would be helpful to describe shortly the differences:mild/moderate and severe (including the Cobbs angle)

2. By using the ODI you decide to measure quality of life (QoL). I would strongly recommend to mention this aspect in your aims.

Discussion: You only discuss one questionnaire but you do use 3- why? By measuring QoL for example by ODI, what did you find? Did you expect it? Are you the first group to measure QoL? Same with GCQ- is there another study to compare your results?

Conclusion: it is rather a summary of the results and a repetition than a conclusion! What was the most striking finding? What can we assume based on your results? From the point of the reader I would prefer to read a message for everyday use (a new aspect!)
